# Proportion of asymptomatic infection among COVID-19 positive persons and their transmission potential: A systematic review and meta-analysis

**Mercedes Yanes-Lane**[1], **Nicholas Winters**[2], **Federica Fregonese**[1], **Mayara Bastos**[1], **Sara Perlman-Arrow**[1], **Jonathon R. Campbell**[1,2,3☯], **Dick Menzies**[1,2,3,4☯] *

1 Research Institute, McGill University Health Centre, Montreal, Quebec, Canada, 2 Department of Epidemiology, Biostatistics and Occupational Health, McGill University, Quebec, Canada, 3 McGill International TB Centre, Montreal, Quebec, Canada, 4 Division of Respiratory Medicine, Department of Medicine, McGill University, Quebec, Canada

☯ These authors contributed equally to this work.
* dick.menzies@mcgill.ca

**Data Availability Statement:** All relevant data are within the manuscript and its Supporting Information files.

## Abstract

### Background

The study objective was to conduct a systematic review and meta-analysis on the proportion of asymptomatic infection among coronavirus disease 2019 (COVID-19) positive persons and their transmission potential.

### Methods

We searched Embase, Medline, bioRxiv, and medRxiv up to 22 June 2020. We included cohorts or cross-sectional studies which systematically tested populations regardless of symptoms for COVID-19, or case series of any size reporting contact investigations of asymptomatic index patients. Two reviewers independently extracted data and assessed quality using pre-specified criteria. Only moderate/high quality studies were included. The main outcomes were proportion of asymptomatic infection among COVID-19 positive persons at testing and through follow-up, and secondary attack rate among close contacts of asymptomatic index patients. A qualitative synthesis was performed. Where appropriate, data were pooled using random effects meta-analysis to estimate proportions and 95% confidence intervals (95% CI).

### Results

Of 6,137 identified studies, 71 underwent quality assessment after full text review, and 28 were high/moderate quality and were included. In two general population studies, the proportion of asymptomatic COVID-19 infection at time of testing was 20% and 75%, respectively; among three studies in contacts it was 8.2% to 50%. In meta-analysis, the proportion (95% CI) of asymptomatic COVID-19 infection in obstetric patients was 95% (45% to 100%) of which 59% (49% to 68%) remained asymptomatic through follow-up; among nursing home residents, the proportion was 54% (42% to 65%) of which 28% (13% to 50%)

**Funding:** This work was funded by McGill Interdisciplinary Initiative in Infection and Immunity (M[i]4) (ECRF-R1-30), which also supports the salary of MYL and SPA. JRC (Award #258907, Award #287869) is supported by a postdoctoral fellowship from the Fonds de Recherche du Québec—Santé. NW (Award #284837) is funded by a doctoral fellowship from the Fonds de Recherche du Québec—Santé. MB and FF are supported through a Canadian Institutes of Health Research grant (FRD331745). The funders had no role in the design and conduct of the study; collection, management, analysis, and interpretation of the data; preparation, review, or approval of the manuscript; and decision to submit the manuscript for publication.

**Competing interests:** We declare no competing interests or conflicts of interest.

remained asymptomatic through follow-up. Transmission studies were too heterogenous to meta-analyse. Among five transmission studies, 18 of 96 (18.8%) close contacts exposed to asymptomatic index patients were COVID-19 positive.

## Conclusions

Despite study heterogeneity, the proportion of asymptomatic infection among COVID-19 positive persons appears high and transmission potential seems substantial. To further our understanding, high quality studies in representative general population samples are required.

## Background

Since December 2019, severe acute respiratory syndrome coronavirus 2 (SARS-CoV-2) has rapidly spread worldwide. Many countries implemented unprecedented measures to control SARS-CoV-2. National lockdowns, physical distancing, quarantine, and travel restrictions were widely implemented. For many countries these measures were successful in controlling the initial wave of the epidemic. However, the disease caused by SARS-CoV-2, coronavirus disease 2019 (COVID-19), can range from asymptomatic infection to severe pneumonia and death [1–4]. The possibility of transmission occurring within this wide presentation range has made sustained control of the disease difficult [5]. Indeed, several early instances of countries easing restrictions and reopening economies and schools have resulted in epidemic recrudescence [6–8].

As more jurisdictions move towards lifting restrictions, public health strategies addressing the spectrum of COVID-19 will be necessary to maintain epidemic control. Persons with asymptomatic COVID-19 infection present a unique challenge as they lack characteristics that might indicate they are infected. Virological studies [9, 10] indicate asymptomatic persons shed similar quantities of virus to symptomatic persons and observational studies have found that younger patients are less likely to present with severe forms of the disease [11, 12]. However, the proportion of infections that are asymptomatic and their infectiousness is still uncertain. Therefore, improving our understanding of the role of persons with asymptomatic COVID-19 infection in the epidemic will be crucial to informing public health strategies.

We conducted a systematic review and meta-analysis to critically evaluate the literature on the proportion of asymptomatic infection among COVID-19 positive persons and their transmission potential.

## Methods

This systematic review adheres to the PRISMA guidelines and our protocol was prospectively registered with PROSPERO (CRD42020181543) [13]. As our scientific understanding of COVID-19 evolved in the process of conducting this review and more data became available, we submitted protocol amendments to our initial strategy for data synthesis. These were submitted to PROSPERO.

### Search strategy and selection criteria

We conducted a systematic review of peer-reviewed or pre-print articles up to June 22, 2020. We designed a search strategy in MEDLINE and EMBASE to identify studies reporting the

proportion of persons with asymptomatic COVID-19 infection and/or the number of close contacts of asymptomatic persons who were diagnosed with COVID-19. The complete search strategy was as follows: exp Asymptomatic Diseases/ OR (asymptomatic or "no symptoms").ti, ab,kw. OR (presymptomatic or Pre-Symptomatic).ti,ab,kw OR (symptomatic).ti,ab,kw AND ("Wuhan Coronavirus" or "novel coronavirus" or SARS-CoV-2 or COVID-19 or 2019-nCoV). ti,ab,kw. In addition, we searched the compendium on COVID-19 and SARS-CoV-2 from MedRxiv and BioRxiv for pre-print articles. All titles, abstracts, and full texts were independently assessed by two reviewers (NW, MYL) without language restriction. These same reviewers also searched reference lists of articles and systematic reviews identified in the search for additional studies.

We included studies for quality assessment that: systematically tested individuals for COVID-19 regardless of symptoms; were cross-sectional or cohort (prospective or retrospective) studies that reported the proportion of COVID-19 positive persons who were asymptomatic at time of testing and/or proportion of COVID-19 positive persons that were asymptomatic at time of testing who later developed symptoms; were case series describing contact or outbreak investigations of asymptomatic index patients and; included ≥25 participants tested for COVID-19 (except in case series describing transmission, which could be any size). Studies were excluded if authors used extrapolated data, based outcomes on modelling, did not define the criteria for SARS-CoV-2 testing, or did not define the population eligible for testing.

## Data extraction and quality assessment

Data from all studies eligible for quality assessment were extracted into a pre-defined extraction form (see S1 File for form) by one of three reviewers (MYL, NW, or SPA) and independently verified by a second reviewer. Disagreements were resolved by consensus with two other reviewers (FF and JRC). If multiple studies reported on the same or overlapping cohorts of participants, information was extracted from each individual study to complement the available information. We defined asymptomatic COVID-19 positive persons as those who did not present any symptoms (or any new symptoms, if pre-existing chronic conditions) at the time of SARS-CoV-2 diagnosis. Pre-symptomatic COVID-19 positive persons were those who were asymptomatic at the time of initial SARS-CoV-2 testing but developed symptoms during study follow-up. Two independent reviewers (MYL, SPA) assessed the quality of each study; any disagreements were resolved by consensus with a third reviewer (JRC). The quality of included studies was evaluated using adapted criteria from the Newcastle-Ottawa scale for cohort studies, and from the Joanna Briggs Institute Prevalence Critical Appraisal Tool for cross-sectional [14, 15]. The quality of included studies was assessed in the domains of selection bias, reporting bias and detection bias (depending on follow-up and outcome measures). We developed our own quality assessment tool for case series exclusively reporting on asymptomatic transmission as we could not identify a validated tool. Our quality assessment tool assessed the domains of reporting bias, detection bias (for contact identification and diagnosis), and misclassification bias (for direction of transmission from the index patient). Signalling questions and domains were selected based on epidemiological knowledge. All studies were classified as low, moderate, or high quality based on presence of bias in each, following grading scales developed *a priori* (see S2 File for detail on quality assessment tools and grading scales).

## Outcomes

There were three primary outcome measures: 1) the proportion of asymptomatic COVID-19 infection among persons testing positive for COVID-19; 2) the proportion of COVID-19

infection that remains asymptomatic throughout study follow-up; and 3) the secondary COVID-19 attack rate among close contacts (both household and non-household) of asymptomatic index patients.

## Data analysis

Because of significant bias concerns, we excluded all low-quality studies from data synthesis and analysis. For all included studies, a qualitative synthesis was performed describing each of the primary outcome measures among different populations included in the studies. To facilitate synthesis, we used the following population categories: general population, contacts, and other populations (this includes healthcare workers in settings other than nursing homes, obstetric patients presenting to hospitals, liver transplant patients presenting to hospitals, persons in congregate settings, patients and staff in nursing homes, and public facing workers). Crude proportions of asymptomatic infection at COVID-19 initial testing, and of COVID-19 infection that remained asymptomatic throughout follow-up, were calculated using n/N based on data availability within each study.

Where at least three studies were conducted in the same population and we judged studies were sufficiently homogenous based on study design and inclusion criteria, we conducted meta-analysis. For overlapping cohorts of participants, the study with the longest study duration or the most complete information on participants was included in the meta-analysis. All meta-analyses were performed with package meta and metaprop function (version 4.12.0) in R (version 3.6.0). For each primary outcome measure, we logit transformed individual study outcomes and applied random-effects meta-analysis using generalized linear mixed models for each as well as for the overall proportion; pooled and individual study proportions were then back-transformed. For all meta-analyses, heterogeneity was quantified using the $I^2$ statistic. In order to determine the proportion of truly asymptomatic individuals (i.e. those who do not develop symptoms at any time during follow up) the total number of COVID-19 infected persons that remain asymptomatic through follow up was used as numerator, and the total number of COVID-19 infections was used as denominator. For studies on transmission, meta-analysis was not performed. For individual transmission studies, we calculated the proportion of contacts traced and tested who were positive for COVID-19 and corresponding exact confidence intervals using the Clopper-Pearson method [16] and report the secondary attack rate overall.

## Patient and public involvement

Patients were not involved in the development of the research question or its outcome measures, conduct of the research, or preparation of the manuscript.

## Results

We identified 6,137 studies in our search and 282 studies entered full text assessment. Of these, 71 studies were included in quality assessment and 28 (39.4%) were high or moderate quality and ultimately included in this review (Fig 1).

Among the 43 low quality studies excluded, 28 were studies on proportion of asymptomatic infection [17–44] and 15 were transmission studies [45–59]. Among the studies on proportion of asymptomatic infection excluded, potential selection bias (21/28; 75%) and detection bias (16/28; 57.1%) were the most common concerns, while for the transmission studies excluded, detection bias (15/15; 100%) and reporting bias (14/15; 93.3%) were the most common concerns (S1 Table). Primary outcome measures extracted from excluded studies are summarized in **S2**–S4 Tables.

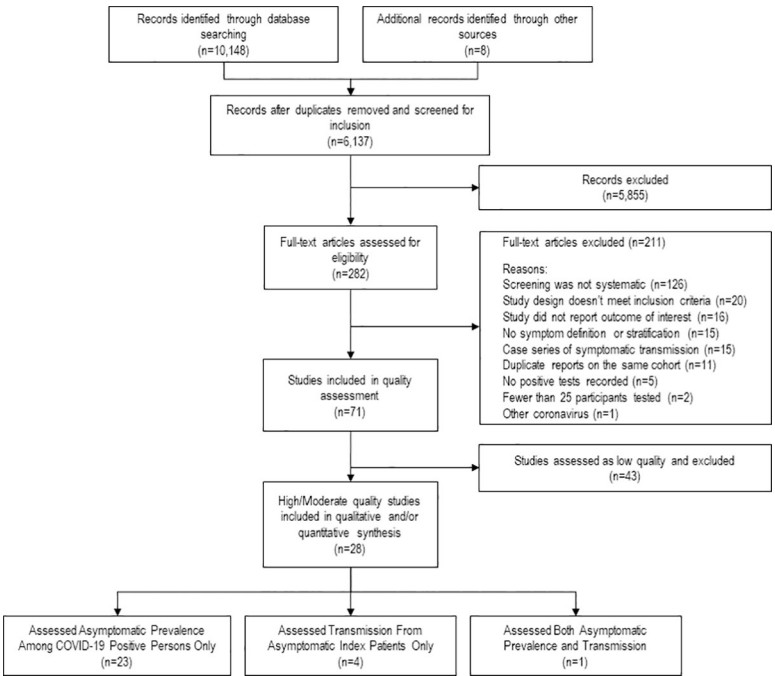

**Fig 1. PRISMA flow diagram of identified and included studies with reasons for exclusion at the full text stage.**

From the 28 high or moderate quality studies included, 24 reported on the proportion of asymptomatic COVID-19 infection at initial testing and/or the proportion of COVID-19 infections remaining asymptomatic throughout follow-up [9, 60–81] and five reported on transmission of COVID-19 from asymptomatic or pre-symptomatic index patients (one study reported both) [82–86].

## Proportion of asymptomatic COVID-19 infection at initial testing & proportion asymptomatic throughout follow-up

Overall, 22 unique cohorts of participants described in 24 studies (15 cohorts, 7 cross sectional) reported the proportion of asymptomatic infection at initial testing and/or the proportion of COVID-19 infections remaining asymptomatic throughout follow-up; study characteristics are summarized in Table 1. Study cohorts were from the USA (n = 10) [6, 9, 60–62, 66, 69, 73, 75, 78, 80, 81], Europe (n = 8) [63, 67, 68, 70, 72, 74, 76, 77], and Asia (n = 4) [65, 70, 79, 84]. Definition of asymptomatic infection was variable among studies, ranging from absence of symptoms in the previous 14 days to only absence of symptoms at time of testing. All studies used reverse transcriptase polymerase chain reaction (RT-PCR) on pharyngeal swabs to diagnose COVID-19. Two cohorts reported on general population samples, three cohorts reported on COVID-19 contacts, and the remaining 17 cohorts reported on other populations, most commonly obstetric patients presenting to hospitals (n = 5) [60, 69, 71, 78, 80, 81] and residents/staff in nursing homes (n = 5) [9, 62–64, 73, 75].

The proportion of asymptomatic COVID-19 infection by population group are reported in Table 2. Among the populations, the median (range) number of people tested for COVID-19 was 118 (34 to 8,437) and the median (range) prevalence of COVID-19 was 8.7% (0.3% to 49%). Of those testing positive for COVID-19, the proportion of asymptomatic infection at initial testing among them ranged from 20% to 75% in the general population (n = 2); 8.2% to

**Table 1. Characteristics of studies reporting proportion of asymptomatic infection among persons positive for COVID-19, by study population (i.e. contacts, general population, and other populations).** No studies were blinded for participants or assessors.

| First Author, Country | Type of study | Inclusion criteria | Definition of asymptomatic | Duration of follow-up from 1st testing (Days) |
|---|---|---|---|---|
| **General Population** | | | | |
| Lavezzo, E. Italy | Cohort | The entire population of the town of Vò, in lockdown from 23 February to 8 March after 1st COVID-19 was reported on 21 February. | Not requiring hospitalization and/or did not report fever (yes/no or a temperature above 37˚C) and/or cough and/or at least two of the following symptoms: sore throat, headache, diarrhoea, vomit, asthenia, muscle pain, joint pain, loss of taste or smell, or shortness of breath. | Range: 7 to 13[2] |
| Snoeck, Ch. Luxembourg* | Cohort | Random selection of adult (>18 years old) population of Luxembourg, stratified by age, gender and electoral districts (2000 participants invited from 18,000 panel members, 1,840 accepted) | None of the symptoms listed in self-reported online questionnaire for the 2 weeks prior to test and in the follow-up | Range: 13 to 37 |
| **Contacts** | | | | |
| Park, S. South Korea | Cohort | All occupants (workers and residents) of a building, closed on 9 March 2020, immediately after an outbreak of COVID-19 was reported. | No symptoms. | For all: 14[1] |
| Schwierzeck, V. Germany | Cohort | HCWs, patients and their accompanying person with contact to SARS-CoV-2 infected individuals (exposure assessed with a risk-based questionnaire), in the Paediatric Dialysis Unit of University Hospital of Munster. | None of the typical COVID-19 symptoms | For all: 10 |
| Zhang, J. China | Cross sectional | All close contacts of an index patient (supermarket employee) and all persons who visited the ZH supermarket tested in the week prior to the index patients last day working (January 16–30, 2020), in Shandong province. | No symptoms | Not followed |
| **Other Populations** | | | | |
| **Healthcare Workers in Settings Other Than Nursing Homes** | | | | |
| Lai, X. China | Cross-sectional | 335 randomly selected HCWs of Fever clinic and of other Departments of Tonji Hospital, Wuhan | None of the following: fever, myalgia or fatigue, cough, sore throat, muscle ache, diarrhoea, headache, dyspnoea, dizziness, sputum production, nausea and vomiting, haemoptysis | Not followed |
| Lombardi, A. Italy | Cohort | All the consecutive HCWs who were tested at the Ca' Granda Ospedale Maggiore Policlinico in Milan, February 24 to March 31, 2020 | None of the following in the 14 days preceding the test: fever, cough, dyspnoea, asthenia, myalgia, coryza, sore throat, headache, ageusia or dysgeusia, anosmia or parosmia, ocular symptoms, diarrhoea, nausea, and vomit. | Range: 52 to 88 |
| Romao, V.C. Portugal* | Cohort | All staff (symptomatic/asymptomatic) of rheumatology department, Hospital de Santa Maria, Lisbon, tested on 15–16 March 2020. | None of the following: fever, cough, dyspnoea, chest tightness, malaise, fatigue, headache, rhinorrhoea, sore throat, anosmia, dysgeusia, arthralgia, myalgia, nausea/vomiting/diarrhoea, dizziness. | Range: 27 to 51 days |
| **Obstetric Patients Presenting to Hospitals** | | | | |
| Andrikopolou, M. & Sutton, D. USA | Cohort | All women, admitted to the labour unit of the New York–Presbyterian Allen Hospital and Columbia University Irving Medical Centre (March 22 to Apr 19, 2020) for delivery or antepartum/postpartum indications | No Covid-19 specific symptoms at testing | For all: Up to 14 days |
| Bianco, A. USA | Cross-sectional | All women who were scheduled for a planned delivery within the Mount Sinai Health system (April 4 to 15, 2020) | None of the following fever or feel hot, cough, shortness of breath, sore throat, vomiting, diarrhoea, rash. | Not followed |

*(Continued)*

**Table 1.** (Continued)

| First Author, Country | Type of study | Inclusion criteria | Definition of asymptomatic | Duration of follow-up from 1st testing (Days) |
|---|---|---|---|---|
| Goldfarb, I. USA | Cohort | All women admitted to labour and delivery units of 2 academic and 2 community hospitals affiliated with Mass General Brigham Health (April 18 to May 5, 2020) | None of the following: fever -subjective or documented-, new cough, shortness of breath, sore throat, muscle aches, new rhinorrhoea, or new anosmia | Time of hospitalization for delivery |
| London, V. USA | Cohort | All pregnant women admitted to antepartum, labour and delivery units at one tertiary care hospital in Brooklyn, New York (March 15 to April 15, 2020). | None of the following: fever, cough, shortness of breath, sore throat, nausea, vomiting | For all: 6 |
| Ochiai, D. Japan | Cohort | All obstetric patients admitted for delivery during universal screening admitted to Keio University Hospital, Tokyo (April 6 to April 27, 2020) | No symptoms of COVID-19 | Range: 7 to 13 |
| **Liver Transplant Patients Presenting to Hospitals** | | | | |
| Ossami, R. Germany | Cohort | All liver transplant outpatients visiting clinic, Berlin, tested between 23 March and 23 April 2020. | None of the following: cough, fever, sore throat, dyspnoea, new/changed sputum, new fatigue, exhaustion, new onset of diarrhoea. | For all: 14 |
| **Congregate Setting** | | | | |
| Baggett, T. USA | Cross sectional | All adults residing in one homeless shelter in Boston on 2 April and 3 April 2020 (excluded if previously tested for COVID19). | No symptoms at testing. | Not followed |
| Ly, T. France* | Cross-sectional | Homeless people, people living in precarious condition and asylum-seekers residing in four shelters, four hotels, and three residences, as well as the employees of these centres, in Marseille (March 26 to April 17,2020) | No fever or respiratory symptoms (cough, rhinorrhoea, dyspnoea, sore throat) | Not followed |
| **Nursing Homes** | | | | |
| Dora, A. USA | Cohort | All residents at skilled nursing facility (Veteran Affairs Greater Los Angeles Healthcare System), regardless of symptoms (March 29 to April 23,2020)- Note: the testing was serial (approximately weekly). | None of the following at time of test of during follow-up, from retrospective notes review: fever, myalgia, headache, cough, dyspnoea, nausea, emesis, diarrhoea, poor appetite. | Range: 14 to 26[4] |
| Graham, N. UK | Cohort | All residents of four nursing homes in central London, tested 15 April to 1 May 2020 | None of the following (from note review): new fever, cough and/or breathlessness; newly altered mental status or behaviour, anorexia, diarrhoea or vomiting. | Up to: 7 |
| Kimball, A. & Arons, M. USA[3] | Serial cross-sectional | All residents on 13 March 2020 at a skilled nursing facility in King County, Washington, after a case of COVID-19 was reported on 1 March 2020. | No symptoms or only stable chronic symptoms (e.g., chronic cough without worsening). | For all: 7 |
| Patel, M., USA | Cohort | Population 1: All residents of Facility where there had been a patient of COVID-19, regardless of symptoms. Population 2: All staff members who worked on the ward where the index case lived. | No symptoms at testing and during follow-up. | For all: 30 |
| Roxby, A. USA | Cohort | All residents and staff at an assisted living community in Seattle after notification of two COVID-19 cases from 5 March to 9 March 2020. First round of testing was carried out on 10 March. | No symptoms at the time of testing or 14 days prior. | For all: 20 |
| **Public Facing Workers** | | | | |

(Continued)

**Table 1.** (Continued)

| First Author, Country | Type of study | Inclusion criteria | Definition of asymptomatic | Duration of follow-up from 1st testing (Days) |
|---|---|---|---|---|
| Lan, F.  USA* | Cross-sectional | All workers older than 18-year-old sent by a grocery retail store for city mandated group-testing in the greater Boston area. | None of the following: fever/chills, headache, running nose, sore throat, cough (acute, new onset, dry or productive), shortness of breath loss of taste or smell, diffuse body ache, fatigue/feeling run down, nausea, diarrhoea | Not followed |

**Abbreviations:** IQR, Interquartile range; HD, hemodialysis, HCW, healthcare workers

*Pre-print studies.

[1]In Park S et al. time in quarantine (not from diagnosis).

[2]In Lavezzo et al. 1st test done 21 to 29 Feb 2020; 2nd test done 7 March 2020.

[3]Same cohort of patients, reported in different studies.

[4]Reviewed March 26 to April 20.

50% in contacts (n = 3); 21.4% to 100% in healthcare workers in settings other than nursing homes (n = 3); 45% to 100% in obstetric patients presenting to hospitals (n = 5); 42.9% to 66.7% among nursing home residents (n = 5); 0% to 50% among nursing home staff (n = 3 studies); and 51% to 87.8% in congregate settings (n = 2). Other populations in which only one study assessed proportion of asymptomatic infection at initial testing included liver transplant patients presenting to hospitals (100%) and public facing workers (76.2%). Within each population group, the proportion of asymptomatic infection did not appear to vary significantly with the number of people tested or the number of people who were COVID-19 positive.

The proportion of COVID-19 positive persons remaining asymptomatic throughout follow-up is described in Table 3. For all but one study, which had follow-up time defined by time in hospital [81], follow-up for symptom development occurred for a minimum of 7 days. Among the general population, one study [67] found that 39.7% remained asymptomatic in the first round of testing and 62.5% in the second round. Among one study [84] in contacts, 4.1% of infections remained asymptomatic. For studies in healthcare workers in settings other than nursing homes, infections remained asymptomatic in 12.2% to 14.3% (n = 2). Among obstetric patients presenting to hospitals, 45% to 100% of infections remained asymptomatic (n = 3), while for residents of nursing homes, 4.3% to 48.1% of infections remained asymptomatic (n = 4). Time to symptom onset was variable and was not reported in five studies. When reported, most studies reported symptoms developing within the first week.

Data was sufficient (i.e., minimum three studies) and study designs and inclusion criteria homogenous enough for meta-analysis in three populations: obstetric patients presenting to hospitals, nursing home residents, and nursing home staff (Table 4). Among obstetric patients presenting to hospitals, the pooled proportion of asymptomatic COVID-19 infection at initial testing in five studies was 95.1 (95% CI: 45.1% to 99.8%; $I^2 = 92\%$), and the proportion of infections remaining asymptomatic throughout follow-up in three studies was 58.8% (95% CI: 48.8% to 68.1%; $I^2 = 0\%$). For nursing home residents, the pooled proportion of asymptomatic infection at initial testing in five studies was 53.6% (95% CI: 42.0% to 64.7%; $I^2 = 40\%$), and the proportion of infections remaining asymptomatic throughout follow-up was 27.9% (95% CI: 13.0% to 49.8%; $I^2 = 84\%$). Among nursing home staff, data was only available to estimate the proportion of asymptomatic infection at initial testing and in four studies this was 46.9% (95% CI: 30.6% to 63.0%; $I^2 = 0\%$).

**Table 2. Population characteristics, COVID-19 prevalence, and proportion of asymptomatic infection among COVID-19 positive persons at time of testing.**

| First Author, Country | Male Sex | Age | Percent of eligible population who were tested | COVID-19 positive and number tested | Proportion of asymptomatic infection at COVID-19 initial testing |
|---|---|---|---|---|---|
| | (%) | (Years) | | n/N (%) | %** |
| **General Population Studies** | | | | | |
| Lavezzo, E. | COVID-19 Positive: | Range:0 to 90 | First test: 85.9% | First test: 73 / 2,812 (2.6) | First test: 53.4% |
| Italy | 59.3% | (18.5% <40 yrs) | Second test: 71.5% | Second test: 8 / 2,322 (0.3)[5] | Second test: 75% |
| Snoeck, Ch. | Tested Population: | Mean (SD): | 92.5% | 5 / 1,842 (0.3) | 20% |
| Luxemburg* | 48.9% | 47 (15) | | | |
| **Contacts** | | | | | |
| Park, S. | Tested Population: | Mean (Range): | 99% | 97 / 1,143 (8.5) | 8.2% |
| South Korea | 27.7% | 38 (20 to 80) | | | |
| Schwierzeck, V. | Tested Population: | Mean: 46 for HCWs, 10 for patients and 32 for accompanying persons. | Not Reported | 12 / 48 (25.0) | 50% |
| Germany | 31% | | | | |
| Zhang, J. | COVID-19 Positive: | 60%: 20–49 yrs old [2] | Not Reported | 25 / 8,437 (0.3) | 12.0% |
| China | 36% | | | | |
| **Other Populations** | | | | | |
| **Healthcare Workers in Settings Other Than Nursing Homes** | | | | | |
| Lai, X. | Not reported | Not reported | 100% | 3 / 335 (0.9) | 100% |
| China | | | | | |
| Lombardi, A. | COVID-19 Positive: | 90% <60yrs old | Not Reported | 139 / 1,573 (8.8) | 28% |
| Italy | 39.9% | | | | |
| Romao, V.C. | COVID-19 Positive: | Mean (SD): | Not Reported | 14 / 34 (41.2) | 21.4% |
| Portugal* | 28.6% | 40(14) | | | |
| **Obstetric Patients Presenting to Hospitals** | | | | | |
| Andrikopolou M. & Sutton, D. | 0% | Not Reported | Not Reported | 75 (Denominator not available) | 84% |
| USA | | | | | |
| Bianco A. | 0% | Mean (SD): | 98.1% | 24 / 155 (15.5) | 100% |
| USA | | 32.7 (6.4) | | | |
| Goldfarb I. | 0% | Not Reported | 99.2% | 20 / 757 (2.6) | 45% |
| USA | | | | | |
| London V. | 0% | Median (IQR): | Not Reported | 10 / 75 (13.3) | 100% |
| USA | | 30.5 (24.5–34.8)[1] | | | |
| Ochiai D. | 0% | Mean (SD): | 100% | 2 / 52 (3.8) | 100% |
| Japan | | 32.5 (0.5) | | | |
| **Liver Transplant Patients Presenting to Hospitals** | | | | | |
| Ossami R. | Asymptomatic: | Mean: | Not Reported | 3 / 101 (3.0) | 100% |
| Germany | 66.7% | 64 | | | |
| **Congregate Setting** | | | | | |

*(Continued)*

**Table 2.** (Continued)

| First Author, Country | Male Sex | Age | Percent of eligible population who were tested | COVID-19 positive and number tested | Proportion of asymptomatic infection at COVID-19 initial testing |
|---|---|---|---|---|---|
| | (%) | (Years) | | n/N (%) | %** |
| Baggett, T. | COVID-19 Positive: | Mean (SD): | 100% | 147 / 408 (36.0) | 87.8% |
| USA | 84.4% | 53.1 (12.8) [2] | | | |
| Ly T. | Tested Population: | Mean (SD): | 78.9% | 49 / 698 (7.0) | 51% |
| France* | 75.8% | 37.4 (16.9) | | | |
| **Nursing Home Residents** | | | | | |
| Dora A. | 100% | Median: | 100% | 19 / 96 (19.8) | 74% |
| USA | | 75 | | | |
| Graham, N. | 37.6% | Median: | 79.4% | 126 / 313 (40.3) | 42.9% |
| UK | | 83 | | | |
| Kimball, A. & Arons, M. | 30.4% [4] | Mean: | First test: 91% [2] | First test: 23 / 75 (31.6) | First test: 52.2% |
| USA | (First test only) | 80.7 | Second test: 94% [3] | Second test: 24 / 49 (49.0) | Second test: 62.5% |
| Patel M., | COVID-19 Positive: | Median: | 99.2% | 27 / 118 (22.9) | 51.9% |
| USA. | 31.4% | 82 | | | |
| Roxby, A. | COVID-19 Positive: | Mean: | 100% | 3 / 80 (3.8) | 66.7% |
| USA | 16.7% [4] | 68.3 [4] | | | |
| **Nursing Home Staff** | | | | | |
| Dora A. | Not reported | Not reported | 100% | 8 / 136 (5.9) | 50% |
| USA | | | | | |
| Patel M., | Not reported | Not reported | 70% | 19 / 42 (45.2) | 41.2% |
| USA. | | | | | |
| Roxby, A. | COVID-19 Positive: | Mean: | 100% | 2/ 62 (3.2) | 0% |
| USA | 0% | 37.5 | | | |
| **Public Facing Workers** | | | | | |
| Lan F. | COVID-19 Positive: | Mean: | 100% | 21 / 104 (20.2) | 76.2% |
| USA* | 53% | 49 | | | |

**Abbreviations**: IQR, Interquartile range; SD, standard deviation; mo, months; yrs, years; HCW, healthcare workers.

*Pre-print studies.

**Proportion was calculated as number of asymptomatic COVID-19 infections at initial testing over COVID-19 positive persons.

[1] In London, et al, age reported is only for asymptomatic COVID-19 positive patients.

[2] In Arons et al: 1st test (March 13th) is done on 76 people: as one person (PCR- on March 13) had already tested positive before March 13, this person is taken out from the denominator (as viral status was previously known).

[3] In Arons et al. 2nd test is done on people testing negative in the 1st test, done one week prior.

[4] Age/Sex among COVID-19 positive patients only.

[5] Excludes people who were positive on the first survey.

## Transmission potential among asymptomatic index patients

Transmission from asymptomatic individuals was assessed in six high or moderate quality studies (4 case series and 1 cohort) [57, 82–86]. The majority of studies were conducted in

**Table 3. Studies reporting the proportion of asymptomatic infections among COVID-19 positive persons at the end of follow-up, and time to symptom onset among those developing symptoms.**

| Author, Country[1] | Proportion of asymptomatic infection among COVID-19 positive persons at initial testing | Proportion of COVID-19 infection that is asymptomatic throughout follow-up | Follow up time after initial testing (days) | Days to symptom onset among those asymptomatic at testing and who developed symptoms during follow up |
|---|---|---|---|---|
| | % (n/N)** | % (n/N) | | |
| **General Population** | | | | |
| Lavezzo, E. | First test: 53.4% (39 / 73) | First test: 39.7% (29 / 73) | Range: | Not reported |
| Italy | Second test: 75% (6 / 8) | Second test: 62.5% (5 / 8) | 7 to 13 | |
| **Contacts** | | | | |
| Park, S. | 8.2% (8 / 97) | 4.1% (4 / 97) | For all: | Maximum: |
| South Korea | | | 14 | 14 |
| **Healthcare Workers in Settings Other Than Nursing Homes** | | | | |
| Lombardi A. | 20.1% (28 / 139) | 12.2% (17 / 139) | Range: | Not reported |
| Italy* | | | 52 to 88 | |
| Romao V.C. | 14.3% (2 / 14) | 14.3% (2 / 14) | Range: | Mean: |
| Portugal* | | | 27 to 51 | 1.5 |
| **Obstetric Patients Presenting to Hospitals** | | | | |
| Andrikopolou M.[2] | 84% (63 / 75) | 61.3% (46 / 75) | For all: | Not reported |
| USA | | | 14 | |
| Goldfarb I. | 45% (9 / 20) | 45% (9 / 20) | Time spent in hospital for delivery | Not reported |
| USA | | | | |
| Ochiai D. | 100% (2 / 2) | 100% (2 / 2) | Range: | No symptoms developed |
| Japan | | | 7 to 13 | |
| **Nursing Homes (Residents Only)** | | | | |
| Kimball, A. & Arons, M. | First Test: 52.2% (12 / 23) | First Test: 4.3% (1 / 23) | For all: | Median (IQR): |
| USA | Second Test: 65.2% (15 / 24) | Second Test: 8.3% (2 / 24) | 7 | 4 (3 to 5) |
| Dora A. | 74% (14 / 19) | 31.6% (6 / 19) | For all: | Range: |
| USA[3] | | | 25 | 1 to 5 |
| Graham, N. | 42.9% (54 / 126) | 38.9% (49 / 126) | Up to: | Not reported |
| UK | | | 7 | |
| Patel M., | 51.9% (14 / 27) | 48.1% (13 / 27) | For all: | Maximum: |
| USA[3] | | | 30 | 8 |

**Abbreviations:** IQR: Interquartile range; NA: Not available in the paper.

* pre-print studies.

**Proportion was calculated as number of asymptomatic COVID-19 infections at initial testing over COVID-19 positive persons.

[1] No studies were blinded to COVID-19 diagnosis. Time between COVID-19 exposure to initial test was not available.

[2] Long follow up of the same cohort reported by Sutton.

[3] Only residents of nursing homes are included, no reports on outcomes of staff tested in these facilities.

China (4/5; 80%) [82, 83, 85, 86], and one study was in South Korea [84]. Study characteristics are reported in S4 Table. Two studies' (40%) contact investigations were exclusively in household contacts, while the remaining three studies' (60%) contact investigations included other close contacts (e.g., work contacts, social contacts). Each of the five included studies reported on index patients who ended up being pre-symptomatic but were asymptomatic during

**Table 4. Pooled estimates of the proportion of asymptomatic infection at initial testing for COVID-19 and proportion asymptomatic at the end of follow-up.**

| Author | Proportion of Asymptomatic Infection at Initial Testing | | | Asymptomatic Infection Throughout Follow-up | | |
|---|---|---|---|---|---|---|
| | Asymptomatic COVID-19 Positive / Total COVID-19 Positive | Proportion at Testing (95% CI) | $I^2$ | Remained Asymptomatic Through Follow-up / Total COVID-19 Positive | Proportion Asymptomatic at End of Follow-up (95% CI) | $I^2$ |
| **Obstetric Patients Presenting to Hospitals** | | | | | | |
| Ochiai, D. | 2/2 | 100.0% (15.8% to 100.0%) | — | 2/2 | 100.0% (15.8% to 100.0%) | — |
| Goldfarb, I. | 9/20 | 45.0% (23.1% to 68.5%) | — | 9/20 | 45.0% (23.1% to 68.5%) | — |
| Andrikopolou, M. | 63/75 | 84.0% (73.7% to 91.4%) | — | 46/75 | 61.3% (49.4% to 72.4%) | — |
| London, V. | 10/10 | 100.0% (69.2% to 100.0%) | — | — | — | — |
| Bianco, A. | 24/24 | 100.0% (85.7% to 100.0%) | — | — | — | — |
| **Pooled estimate** | **108/131** | **95.1% (45.1% to 99.8)** | **92%** | **74/97** | **58.8% (48.8% to 68.1%)** | **0%** |
| **Nursing Home Residents** | | | | | | |
| Patel, M. | 14/27 | 51.9% (31.9% to 71.3%) | — | 13/27 | 48.1% (28.7% to 68.1%) | — |
| Dora, A. | 14/19 | 73.6% (48.8% to 90.8%) | — | 6/19 | 31.6% (12.61% to 56.6%) | — |
| Graham, N. | 54/126 | 42.9% (34.1 to 51.9%) | — | 49/126 | 38.9% (30.3% to 47.9%) | — |
| Aarons, M. | 27/47 | 57.4% (42.1% to 71.7%) | — | 3/47 | 6.4% (1.3% to 17.5%) | — |
| Roxby, A. | 2/3 | 66.7% (9.4% to 99.2%) | — | — | — | — |
| **Pooled estimate** | **111/222** | **53.6% (42.0% to 64.7%)** | **40%** | **71/220** | **27.9% (13.0% to 49.8%)** | **84%** |
| **Nursing Home Staff** | | | | | | |
| Patel, M. | 8/19 | 42.1% (20.3% to 66.5%) | — | — | — | — |
| Dora, A. | 4/8 | 50.0% (15.7% to 84.3%) | — | — | — | — |
| Graham, N. | 3/3 | 100.0% (29.2% to 100.0%) | — | — | — | — |
| Roxby, A. | 0/2 | 0.0% (0.0% to 84.2%) | — | — | — | — |
| **Pooled estimate** | **15/32** | **46.9% (30.6% to 63.0%)** | **0%** | **—** | **—** | **—** |

Notes: 95% CI, 95% confidence interval; proportions are calculated using a logit transformation of a random effects meta-analysis.

contact, while one study also reported on index patients who remained asymptomatic throughout infection. Data on time to testing among contacts and time to symptom onset are provided in S5 Table. Overall, the five studies included 13 index patients who had 96 contacts traced and tested, with 18 (18.8%) being positive for COVID-19.

Secondary attack rates ranged from 0% to 80% among the studies (Table 5). For index patients who were pre-symptomatic, 18 of 92 (19.6%) contacts who were exposed while index patients were asymptomatic tested positive for COVID-19. In the one study that also reported on index patients who remained asymptomatic throughout infection, none of the four exposed contacts tested positive for COVID-19.

**Table 5. Pooled estimates of secondary attack rates, only high and moderate quality studies.**

| First Author | Number of Index Patients | Type of Contacts Traced in Study | Contacts | | Secondary Attack Rate |
|---|---|---|---|---|---|
| | | | Number of Contacts Tested | Number of Contacts Testing Positive | (95% CI)† |
| Park, S. | 4˚ | Household Contacts Only | 11 | 0 | 0% |
| | | | | | (0% to 28.5%) |
| Park, S. | 4˚˚ | Household Contacts Only | 4 | 0 | 0% |
| | | | | | (0% to 60.2%) |
| Ye, F. | 1˚ | Close Contacts | 44 | 4 | 9.1% |
| | | | | | (2.5% to 21.7%) |
| Huang, L. | 1˚ | Close Contacts | 22 | 7 | 31.8% |
| | | | | | (13.9% to 54.9%) |
| Li, P. | 1˚ | Household Contacts Only | 5 | 4 | 80% |
| | | | | | (28.4% to 99.5%) |
| Xiao, W. | 2˚ | Household and Close Contacts | 10 | 3 | 30% |
| | | | | | (6.7% to 65.2%) |
| **Simple Pooled Estimate** | **13** | | **96** | **18** | **18.8%*** |

Abbreviations: 95% CI, 95% confidence interval.

*Estimates have been simply pooled to facilitate interpretation of the body of evidence. Since studies are too heterogeneous in methods of contact tracing, confidence intervals on these estimates have not been calculated to not overstate any sense of precision.

˚Index patients were pre-symptomatic (exposure occurred prior to symptom onset).

˚˚Index patients remained asymptomatic.

†Confidence intervals calculated for individual studies only, using the Clopper-Pearson exact method.

## Discussion

In this systematic review and meta-analysis, we found that the proportion of asymptomatic infections at initial testing for COVID-19 appears high in many populations and such persons may have substantial transmission potential. Given the variability in study designs and settings and the scarcity of high-quality studies for different populations, pooled estimates could only be calculated for few populations. These included obstetric patients and residents and staff of nursing homes, population groups with unique characteristics that may not be generalizable to the general population. Therefore, caution must be applied when trying to estimate a precise number for the proportion of COVID-19 infections asymptomatic at initial testing and the overall proportion of infections that will remain asymptomatic.

Most studies included in this systematic review reported on relatively small cohorts of people (<100) who were COVID-19 positive, which may limit the precision of estimates. In a study that tested almost all residents of a municipality during the initial wave of the epidemic in Italy [67], approximately half of all participants with COVID-19 were asymptomatic at testing and by the end of follow-up approximately 40% remained asymptomatic. This is similar to the proportion of infections that were asymptomatic estimated by seroprevalence surveys. Surveys performed in both Italy [87] and Spain [88] estimated that approximately one-third of seropositive participants had previous asymptomatic infections, although such classifications could be affected by symptom-recall bias.

Few thorough case-series were identified reporting transmission from asymptomatic persons and among the five studies included, most traced and tested limited contacts and only one included index patients who were asymptomatic throughout infection. While it is

understandable that in the first months of the pandemic any case-series are of great interest, there is a limited value to the evidence this type of research presents. In order to provide a higher level of evidence, future COVID-19 research should focus on using cohort study designs that include: systematic screening, clear reporting of participant selection criteria, ascertainment of time of exposure, time from exposure to diagnosis, adequate follow-up time after diagnosis, assessment of time to symptom onset, and time to RT-PCR negativity. Additionally, new phone applications for contact tracing coupled with systematic surveillance surveys could work to identify persons while they are asymptomatic and trace their close contacts to provide more evidence on their role in transmission.

Given heterogeneity between studies, we could not systematically compare proportions of asymptomatic infection in different age categories or by sex. Although a high proportion of persons with asymptomatic COVID-19 infection was estimated in meta-analysis for studies with younger populations (e.g., obstetric patients), it was also high in older age groups (e.g., nursing home residents). However, in these younger populations it appeared fewer people developed symptoms compared to older groups, during similar follow up times. This is in line with reports of higher disease severity among older persons, but must be confirmed in population studies [89].

From included studies, we could conclude that the proportion of asymptomatic infection at initial testing for COVID-19 is not negligible in any population, similar to findings of a narrative review on the topic [90], and likely has an important role in viral transmission. While larger included studies suggest 40–50% of persons asymptomatic at testing did develop symptoms during follow-up, the lag time between diagnosis and symptom onset indicates that if untested, people may unknowingly spread the disease for up to two weeks before a diagnosis based on symptom screening.

The transmission studies in our review documented substantial transmission—like that seen in a large study in South Korea [91]—but data was not available to compare secondary attack rates between pre-symptomatic and asymptomatic index patients. It is likely that transmission from index patients who remained asymptomatic throughout infection may not be detected or reported due the nature of the asymptomatic infection. Therefore, secondary attack rates estimated from these studies may not be truly representative of real-world attack rates, but when combined with other studies on viral shedding [92, 93], provide evidence that asymptomatic persons can readily transmit SARS-CoV-2. In addition, studies have identified high viral loads in asymptomatic persons for up to 9 days, and in pre-symptomatic persons for up to 6 days prior to symptoms [9, 10]. These viral loads are like those found in symptomatic persons [9, 10, 94, 95]. Together, these findings suggest that exclusively carrying out symptom-based testing will not be sufficient to eliminate transmission and will likely miss a large proportion of SARS-CoV-2 infections.

Rapid identification of COVID-19 positive persons, isolation, and contact tracing are essential for detection and prevention of secondary cases. In the absence of symptoms, strategies must be proactive. Testing of high-risk populations such as healthcare workers, workers in long-term care facilities, public facing workers, and people in congregated settings should be conducted at frequencies informed by circulating COVID-19 prevalence to identify asymptomatic infections and interrupt transmission chains. This testing would be facilitated by development and distribution of inexpensive, point of care tests for COVID-19. In symptomatic persons diagnosed with COVID-19, contact tracing should be extended to several days prior to symptom onset (i.e., up to 6 days based on viral shedding) [9, 58] to ensure persons exposed to index patients while they were asymptomatic are identified. Finally, current non-pharmaceutical measures, such as frequent handwashing, physical distancing, and use of facemasks should be continued as they limit exposure to persons who are infected but asymptomatic.

## Strengths and limitations

This systematic review and meta-analysis provides a detailed synthesis of the current and growing literature on the role of asymptomatic persons with COVID-19. We were able to include evidence from several populations and risk groups, which can be used to inform public health practice. By only including studies that tested populations systematically, without pre-selecting symptomatic or asymptomatic populations, we tried to limit the potential for selection bias and thus increase the accuracy of our estimates. We excluded studies assessed to be of low quality, as these did not include information on the population tested, methods for ascertaining the presence of symptoms or definition of asymptomatic, which we deemed to be important for reducing bias. By including studies that had rigorous methodologies as well as complete reporting, we were able to provide more accurate estimates, however, considerations need to be taken regarding the generalizability of results.

This study is not without its limitations. Since studies were highly heterogeneous—in terms of design, follow up time, definition of asymptomatic, setting and population included—we could not carry out meta-analyses for many populations. Studies also differed in terms of when they were conducted in relation to epidemic stage; however, by only including studies with systematic screening, this should overcome potential biases. We could not identify high-quality studies in children and so this important population was not included in this review. Another important limitation is the fact that no tools were identified to evaluate the quality of transmission studies, and although we created a tool for this purpose, it is not validated. It is possible there is publication bias towards case-series documenting transmission from asymptomatic and pre-symptomatic individuals, given that studies in which transmission from asymptomatic individuals was not documented are not available. This may cause us to overestimate the true secondary attack rate from these types of infections. We attempted to mitigate this risk by applying strict criteria for inclusion, which necessitated clear reporting of the contact investigation, number of contacts traced, number of contacts tested, and time of transmission.

## Conclusion and policy implications

Among the populations evaluated, many COVID-19 infections were asymptomatic and transmission in the asymptomatic period was documented in numerous studies. Additional, unbiased research would further help inform the role that asymptomatic infections are playing in the pandemic. Proactive steps should be taken to develop public health strategies aimed to identify and mitigate transmission from asymptomatic individuals. Systematic testing of high-risk populations should be performed regardless of symptoms. This should be augmented with thorough tracing and testing of all contacts in addition to existing non-pharmaceutical interventions. Given the large proportion of COVID-19 infections that are asymptomatic, such multifaceted strategies will be essential to prevent recrudescence as countries ease restrictions and reopen economies and schools.

## Supporting information

**S1 Checklist. PRISMA 2009 checklist.**
(DOC)

**S1 File. Data extraction form.**
(DOCX)

**S2 File. Criterial for quality assessment in included studies.**
(DOCX)

**S1 Table. Quality assessment of all included studies.**
(DOCX)

**S2 Table. Low quality studies reporting population characteristics, COVID-19 prevalence, and proportion of asymptomatic infection among COVID-19 positive persons at time of testing.**
(DOCX)

**S3 Table. Low quality studies reporting proportion of asymptomatic infections among COVID-19 positive persons through follow-up, and time to symptom onset among those developing symptoms in follow-up.**
(DOCX)

**S4 Table. Transmissibility of infection for asymptomatic and pre-symptomatic patients: From studies on contact investigations.**
(DOCX)

**S5 Table. Transmission from asymptomatic/pre-symptomatic index patients to contacts and time to symptoms development in positive contacts (high and moderate quality studies).** All index patients were asymptomatic when they were in contact with others.
(DOCX)

## Acknowledgments

We thank Zhiyi Lan for his help translating and his valuable insights.

## Author Contributions

**Conceptualization:** Mercedes Yanes-Lane, Jonathon R. Campbell, Dick Menzies.

**Data curation:** Nicholas Winters, Federica Fregonese, Sara Perlman-Arrow.

**Formal analysis:** Mercedes Yanes-Lane, Nicholas Winters, Mayara Bastos.

**Funding acquisition:** Jonathon R. Campbell, Dick Menzies.

**Investigation:** Mercedes Yanes-Lane, Nicholas Winters, Federica Fregonese, Sara Perlman-Arrow.

**Methodology:** Mayara Bastos.

**Supervision:** Mercedes Yanes-Lane, Jonathon R. Campbell, Dick Menzies.

**Writing – original draft:** Mercedes Yanes-Lane, Federica Fregonese.

**Writing – review & editing:** Jonathon R. Campbell, Dick Menzies.

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
