## [Decision Letter · Decision Letter 0]

12 Oct 2020

PONE-D-20-29770

Proportion of asymptomatic infection among COVID-19 positive persons and their transmission potential: a systematic review and meta-analysis

PLOS ONE

Dear Dr. Yanes Lane,

Thank you for submitting your manuscript to PLOS ONE. After careful consideration, we feel that it has merit but does not fully meet PLOS ONE’s publication criteria as it currently stands. Therefore, we invite you to submit a revised version of the manuscript that addresses the points raised during the review process.

The reviewers have commented on your above paper. They have suggested that this manuscript be revised according to the reviewers suggestions and resubmitted.  Provided you address the changes recommended, the manuscript will be accepted for publication. 

We look forward to receiving your revised manuscript.

Kind regards,

Prof. Raffaele Serra, M.D., Ph.D

Academic Editor

PLOS ONE

Additional Editor Comments:

The reviewers have commented on your above paper. They have suggested that this manuscript be revised according to the reviewers suggestions and resubmitted.

Journal Requirements:

2. In the Discussion section, please discuss how results can be interpreted given the quality of the included studies.

Reviewers' comments:

Reviewer's Responses to Questions

**Comments to the Author**

1. Is the manuscript technically sound, and do the data support the conclusions?

Reviewer #1: Yes

Reviewer #2: Yes

2. Has the statistical analysis been performed appropriately and rigorously? 

Reviewer #1: Yes

Reviewer #2: Yes

3. Have the authors made all data underlying the findings in their manuscript fully available?

Reviewer #1: Yes

Reviewer #2: Yes

4. Is the manuscript presented in an intelligible fashion and written in standard English?

Reviewer #1: Yes

Reviewer #2: Yes

5. Review Comments to the Author

Reviewer #1: The authors conducted a systematic review and meta-analysis to critically evaluate the literature on the proportion of asymptomatic infection among COVID-19 positive persons and their transmission potential. The article is well written and in my opinion is very informative and important for scientific community. Well done!

Reviewer #2: This systematic review and meta-analysis was rigorously executed and described in this well-written paper. The design was appropriate for the question addressed. Although the review was limited by the heterogeneity of the publications included, the authors were careful to minimize bias by selecting papers that met clearly described criteria, and their conclusions are supported by their findings. I found no major concerns in this paper. The following minor suggestions are offered to strengthen the paper overall.

1. Since the authors mention in the Discussion that data were inadequate to draw conclusions regarding asymptomatic infection and age, a sentence or two in the Introduction summarizing the current state of knowledge regarding symptoms by age would be appropriate.

2. Line 218 - Superscript the "2" in I2.

3. Line 301 - I believe that "COVID-19" should be replaced with "SARS-CoV-2" since the virus is being transmitted, rather than the disease.

4. Line 309 - the list of high risk populations that require asymptomatic screening could also include younger people in congregate settings where safety precautions are not strictly followed (i.e., college students).

6. PLOS authors have the option to publish the peer review history of their article (what does this mean?). If published, this will include your full peer review and any attached files.

Reviewer #1: **Yes: **Nicola Ielapi

Reviewer #2: No

---

## [Author Response · Author response to Decision Letter 0]

16 Oct 2020

PLOS ONE

Dear Dr. Serra, 

Many thanks to you and the reviewers for your time in reviewing our manuscript. We have revised the paper in light of the comments received and provided a point-by-point response, where necessary. We have provided a track changes and a clean version of the manuscript. In our responses below, when we refer to line numbers, these refer to line numbers in the clean version. We appreciate the opportunity to revise our manuscript. 

Yours very truly,

Dick Menzies and Mercedes Yanes Lane, On behalf of all authors 

Editor comments: 

Response: We followed the formatting guidelines and updated as necessary. 

2. In the Discussion section, please discuss how results can be interpreted given the quality of the included studies.

Response: This is a helpful comment. We have added the following text to the discussion “We excluded studies assessed to be of low quality, as these did not include information on the population tested, methods for ascertaining the presence of symptoms or definition of asymptomatic”, which we deemed to be important for reducing bias. By including studies that had rigorous methodologies as well as complete reporting, we were able to provide more accurate estimates, however, considerations need to be taken regarding the generalizability of results.” This revision can be found on lines 323 to 328. 

3. Please include captions for your Supporting Information files at the end of your manuscript, and update any in-text citations to match accordingly.

Response: We have added captions for the Supporting Information to the end of the manuscript. 

Reviewer comments:

1. Since the authors mention in the Discussion that data were inadequate to draw conclusions regarding asymptomatic infection and age, a sentence or two in the Introduction summarizing the current state of knowledge regarding symptoms by age would be appropriate.

Response: The following statement (with the appropriate references) has been added to the introduction “observational studies have found that younger patients are less likely to present with severe forms of the disease.” This revision can be found on lines 49 to 50. 

2. Line 218 - Superscript the "2" in I2.

Response: We have corrected the superscript. 

3. Line 301 - I believe that "COVID-19" should be replaced with "SARS-CoV-2" since the virus is being transmitted, rather than the disease.

Response: Thank you for flagging this. We have changed accordingly.

4. Line 309 - the list of high risk populations that require asymptomatic screening could also include younger people in congregate settings where safety precautions are not strictly followed (i.e., college students).

Response: This is a helpful suggestion. We have updated the list to include populations in congregate settings.

---

## [Editor Report · Decision Letter 1]

19 Oct 2020

Proportion of asymptomatic infection among COVID-19 positive persons and their transmission potential: a systematic review and meta-analysis

PONE-D-20-29770R1

Dear Dr. Yanes Lane,

We’re pleased to inform you that your manuscript has been judged scientifically suitable for publication and will be formally accepted for publication once it meets all outstanding technical requirements.

Kind regards,

Prof. Raffaele Serra, M.D., Ph.D

Academic Editor

PLOS ONE

Additional Editor Comments (optional):

amended manuscript is acceptable
---

## [Editor Report · Acceptance letter]

22 Oct 2020

PONE-D-20-29770R1 

Proportion of asymptomatic infection among COVID-19 positive persons and their transmission potential: a systematic review and meta-analysis 

Dear Dr. Yanes Lane:

I'm pleased to inform you that your manuscript has been deemed suitable for publication in PLOS ONE. Congratulations! Your manuscript is now with our production department. 

Kind regards, 

on behalf of

Prof. Raffaele Serra 

Academic Editor

PLOS ONE